# Gender and racial bias issues in a commercial "tone of voice" analysis system

**Nicole R. Holliday** [1]*, **Paul E. Reed**[2]

**1** Department of Linguistics, University of California, Berkeley, Berkeley, California, United States of America, **2** Department of Communicative Disorders, University of Alabama, Tuscaloosa, Alabama, United States of America

* nicole.holliday@berkeley.edu

## Abstract

Social Feedback Speech Technologies (SFST) are programs and devices, often "AI"-powered, that claim to provide users with feedback about how their speech sounds to other humans. To date, academic research has not focused on how such systems perform for a variety of speakers. In 2020, Amazon released a wearable called *Halo*, touting its fitness and sleep tracking, as well as its ability to evaluate the wearer's voice to help them "understand how they sound to others". The band presents its wearer with 'Positivity' and 'Energy' scores, as well as qualitative evaluations of the voice: adjectives such as 'confident', 'hesitant', 'calm', etc. This study evaluates how Halo performs for American English speakers of different races and genders. We recorded Black and white men and women reading three passages aloud and played them back to the same Halo device in identical positions. We then obtained Halo's Energy and Positivity scores (out of 100), as well as the device's qualitative descriptors of 'tone of voice' for each subject. We subsequently analyzed effects of different acoustic properties, as well as speaker race/gender and the interaction, for how the device scores 'tone of voice'. Overall, Halo's Energy ratings and qualitative descriptors are biased against women and Black speakers. Halo's Positivity scores appear to be based on lexical sentiment analysis and therefore do not vary substantially by speaker. We conclude by discussing the expanding role of SFSTs and their potential harms related to the reinforcement of existing societal and algorithmic biases against marginalized speakers.

## Introduction

Social Feedback Speech Technologies (SFSTs) are a rapidly emerging set of systems in the area of language technology. Systems such as the Amazon Halo, the Zoom Revenue Accelerator, and Poised AI claim to provide actionable feedback for users in order to help them "communicate more effectively" or "understand how they sound to others" [1–5]. However, as such technologies are in their infancy, their ability to provide fair and accurate feedback to users from a variety of backgrounds across different speech conditions may not be fully developed. Additionally, there are inherent challenges in building such systems, given massive variability in both linguistic performance as well as social norms and expectations within and across speaker populations. There are systematic differences

**Data availability statement:** Data cannot be shared publicly because of Pomona College IRB restrictions on PII in audio recordings. The data underlying the results presented in the study are available from Pomona College IRB at irb@pomona.edu.

**Funding:** The author(s) received no specific funding for this work.

**Competing interests:** The authors have declared that no competing interests exist.

between social groups with respect to pronunciation, sentence structure, contextual meaning and other factors that systems would need to consider for effective performance. There are also technical limitations posed by how speakers interact with and use technologies in real-world settings. Nevertheless, the availability and functions of such devices and systems continue to grow across domains. The current paper aims to discuss broad challenges of SFSTs, as well as presents the results of one of the first linguistic studies that investigates issues of bias in the performance of an SFST system, the Amazon Halo, which operated from 2020–2023.

Much of the previous research on potential algorithmic bias in speech technologies has focused on issues of how systems designed for Automatic Speech Recognition (ASR) seem to underperform for speakers from marginalized racial or linguistic backgrounds. Such studies generally focus primarily on differences in Word Error Rate (WER) across populations. In a study of WER and dialect differences, Koenecke et al (2020) find that 5 different ASR systems "exhibited substantial racial disparities, with an average word error rate (WER) of 0.35 for Black speakers compared with 0.19 for white speakers" [6, pg. 1]. With respect to speech recognition, Black speakers and L2 (second language) English speakers often experience systematically degraded performance [7,8]. Dubois et al (2024) also observe substantially worse performance for L2 English speakers across a variety of ASR systems designed for captioning [9]. In general, researchers cite lack of variation and initial rater bias as significant drivers of this unequal performance [10]. We hypothesize that SFSTs are likely to face similar challenges during their design and training phases, especially due to the fact that they rely on ASR as a first step in their evaluation process. Additionally, since nearly all modern speech recognition systems are based on large language models (LLMs), there is no transparency for designers or the public about the contents of their training data, which means that users cannot evaluate how the systems work across different groups and speech contexts. Beyond challenges related to training, however, SFSTs also face a number of other issues related to their core ideal functions.

SFSTs are generally marked to users as helping them understand how they sound to others or how they may be perceived by listeners. The first major challenge with this goal is that such systems generally do not have access to information about the speaker's desired audience, and therefore can only use information from the speaker side of the interaction [11]. Listeners from different backgrounds and with different linguistic experiences often interpret the same signal differently and their evaluation of any given speech-signal is context-dependent [12,13]. Secondly, many SFSTs are marketed for use in a variety of real-world contexts; see for example, Amazon's promotional material for the Halo, which depicts a character in a tense interaction with his presumed domestic partner because he didn't sufficiently "compliment her talents" [1]. Depending upon the ambient noise in the environment, the way that devices and human listeners interpret responses may be deeply influenced by everyday sounds such as traffic, air conditioners, animals, or other voices [14]. While systems are getting better at isolating voices and controlling for ambient noise, the sheer range of contexts in which SFSTs would need to operate to be effective presents a significant challenge for providing consistent feedback across real-world environments. Third, since SFSTs analyze conversational data, the range of sociolinguistic behavior and norms is exponentially broad. Such systems are likely to struggle to interpret common features of discourse that are culturally variable and compositional in meaning, such as sarcasm, jokes, and ritual insults [15]. Compounding these challenges is the fact that the same utterance may be considered comedic in one context, and wildly inappropriate in another, given elements such as the speaker's identity, social relationship and social distance between the interlocutors [16].

In order to begin to observe how SFSTs behave in a real-world environment, with a specific focus on examining performance for different types of speakers, our experiment tests the performance and evaluative capacities of one technology, the Amazon Halo. This device was one of the first widely-available commercial wearables designed to provide feedback about a user's language practices. The Halo was released in Summer 2020 and was designed and marketed as a health and wellness device. Unlike other fitness devices, the Halo included a unique "tone" feature, which marketing by Amazon [3] described in this way:

> "The globally accepted definition of health includes not just physical but also social and emotional well-being. The innovative Tone feature uses machine learning to analyze energy and positivity in a customer's voice so they can better understand how they may sound to others, helping improve their communication and relationships. For example, Tone results may reveal that a difficult work call leads to less positivity in communication with a customer's family, an indication of the impact of stress on emotional well-being".

In materials like this, Amazon stated that the device was designed to improve the user's communication skills and therefore their relationships. The Halo device could be activated to listen to specific speech samples that the user chooses and then to provide Energy and Positivity scores out of 100, as well as qualitative feedback in the form of a list of descriptive adjectives for each phrase in the sample. Amazon did not provide any publicly available information about how the device was trained to make these evaluations and what type of speech and speakers its ASR was trained to evaluate. However, in previous work that found bias against female speakers in the Halo's qualitative evaluations, Yeung et al [17] hypothesize that one source of this bias is likely that the Halo's machine learning (ML) model itself contained biased input data, an issue that has also presented a pervasive challenge for other AI-based evaluation systems.

The current study seeks to address whether there are systematic differences in the Halo scores (Energy and Positivity) for speakers from different racial and gender groups. In order to answer this question, and to determine how Halo evaluates different speaker groups, we designed a laboratory experiment. Due to previous studies, we hypothesized that there could be systematic differences such that female and Black speakers would receive lower scores. We recorded 58 Midwestern American English speakers reading 3 passages each, which we then played to a Halo device in a unique session for each speaker. All speakers self-identified as either male or female and as Black or white, and all were native English speakers with no speech or hearing impairments. After each session, we downloaded Halo's evaluation data directly from Amazon, then reset the device. We subsequently conducted LASSO regression models for Energy and Positivity, which included race, gender, and 37 acoustic properties. Finally, we conducted a qualitative analysis of Halo's descriptive adjectives for the speakers. The analysis reveals significant biases against women and Black speakers with respect to both Energy scores and descriptive adjectives.

## Results

The results of our analyses show differences in the Halo's ratings primarily based on race and gender, with none of the 37 acoustic properties emerging as major contributors to the scores. A LASSO regression model showed that Positivity was minorly influenced by gender and race, with white having the largest positive coefficient (.285) and Women having a positive coefficient (.205). Fig 1 shows a boxplot of Positivity scores separated by race and gender. However, one can see there is much overlap, and the average scores are remarkably close (50.4 for Women and 50.2 for Men).

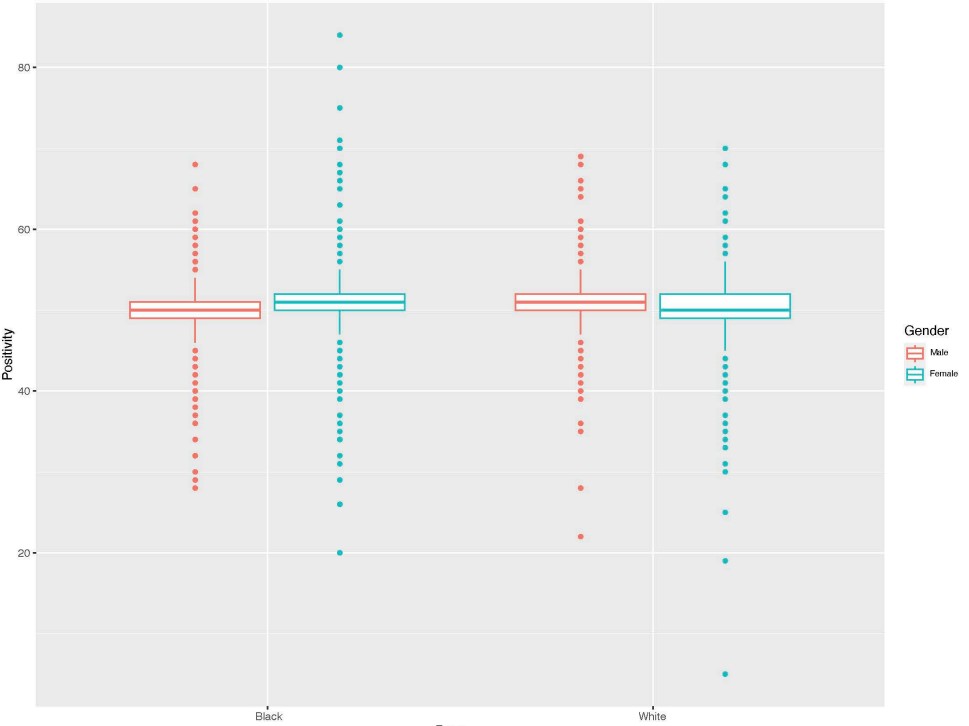

**Fig 1. Boxplot of Positivity scores separated by Race (Black on the left, white on the right) and Gender (red is male, blue is female).**

It is notable that for Positivity, the effect size of a t-test by gender (Cohen's d = 0.06) and Race (Cohen's d = 0.03) both appear to be trivial. The speakers in this data set all read the same passage, so one probable explanation for such small differences is that the Halo may not be evaluating "tone of voice" at all with respect to this variable, given that none of the voice quality variables included in the model showed effects that contributed substantially to score differences (the largest effect was f0 with a coefficient of 0.076). We instead hypothesize that the Halo may use other criteria, such as a speech-to text-based system that conducts subsequent sentiment analysis of the text.

Turning to the Halo's scores for Energy, we see a more complicated picture. The LASSO results demonstrate that race and gender had an impact on Halo's Energy scores. Here we present the three factors with the largest positive and negative coefficients from the model, displayed in Table 1.

Table 1 shows a benefit for white speakers, and a penalty for women speakers. The interaction between f0 and gender shows that a lower f0 for women speakers penalizes them, removing the benefit. When we consider that Black women speakers have a lower f0 than white

**Table 1. Results from the LASSO model displaying the three largest positive and negative coefficients.**

| Factor | Coefficient |
| --- | --- |
| RaceW | .326 |
| GenderF | -.137 |
| GenderF:f0 | -.123 |

women speakers, we see then that such an interaction shows a double penalty. Fig 2 below shows a boxplot of the Energy scores split by race and gender, and these differences emerging from the model become more apparent.

In Fig 2, we see that both Black men (mean = 46.54, sd = 3.49) and Black women (mean = 46.35, sd = 3.93) speakers receive somewhat lower Energy scores than their white counterparts, white men (mean = 47.04, sd = 3.23) and white women (mean = 46.82, sd = 3.45). One can also see that the Black speakers have greater standard deviations in their Energy scores. It is also noteworthy that Black women had the lowest Energy scores overall and as well as the greatest standard deviation of their scores, indicating that the Halo may be less consistent in its scores for this group than others.

We used T-tests to focus our analysis on patterns from the LASSO regression above. The results revealed significant differences between the Energy scores between Black Speakers and white Speakers (t = −145.45, df = 448170, p-value < 2.2e-16). This difference showed a much greater effect size, with a Cohen's d of 0.4, demonstrating a medium effect. T-tests also showed that Black men had lower Energy scores than white Men (t = −63.783, df = 743366, p-value < 2.2e-16), with a moderate effect size (Cohen's d = 0.47). T-tests revealed that Black women had lower Energy scores than white Women (t = −57.265, df = 786212, p-value < 2.2e-16), with a moderate effect size (Cohen's d = 0.37).

Halo also provides qualitative descriptive adjectives for each utterance, which demonstrate a similar pattern of racial and gender biases. Fig 3 presents a word cloud of the descriptors. Each quadrant shows the most common descriptors for each gender and race pair. Descriptors close to the center of the plot are equally spread across groups, e.g., 'interested' was used 17,721 times for utterances from Black men, 21,965 for utterances from Black women, 21,321

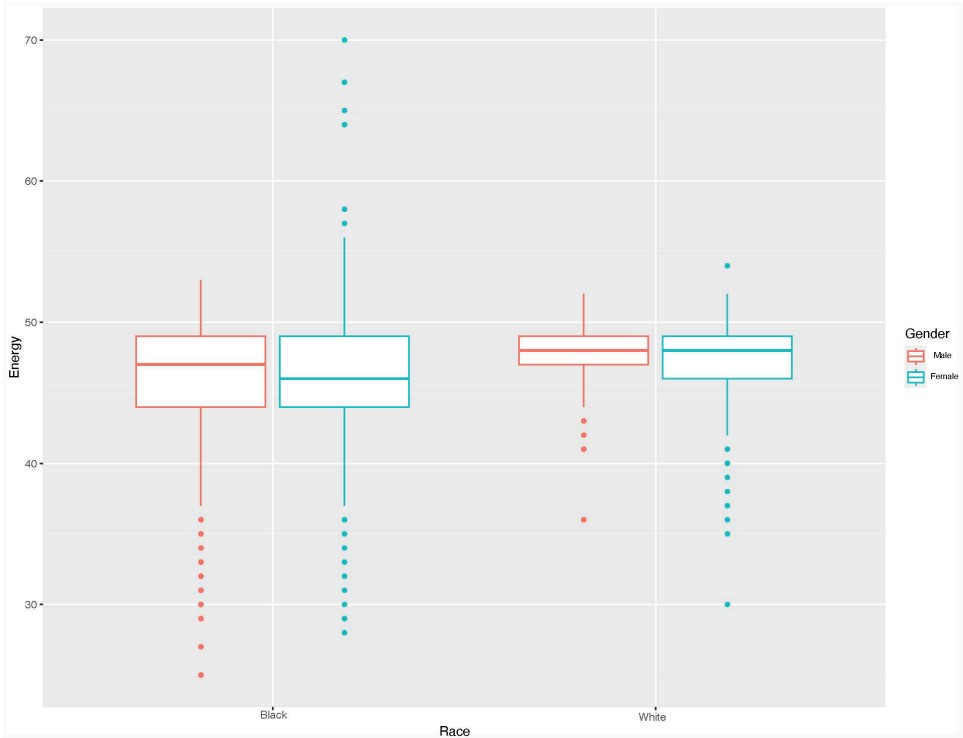

**Fig 2. Boxplot of Energy scores split by race (Black on left, white on right) and gender (male are red, female are blue).**

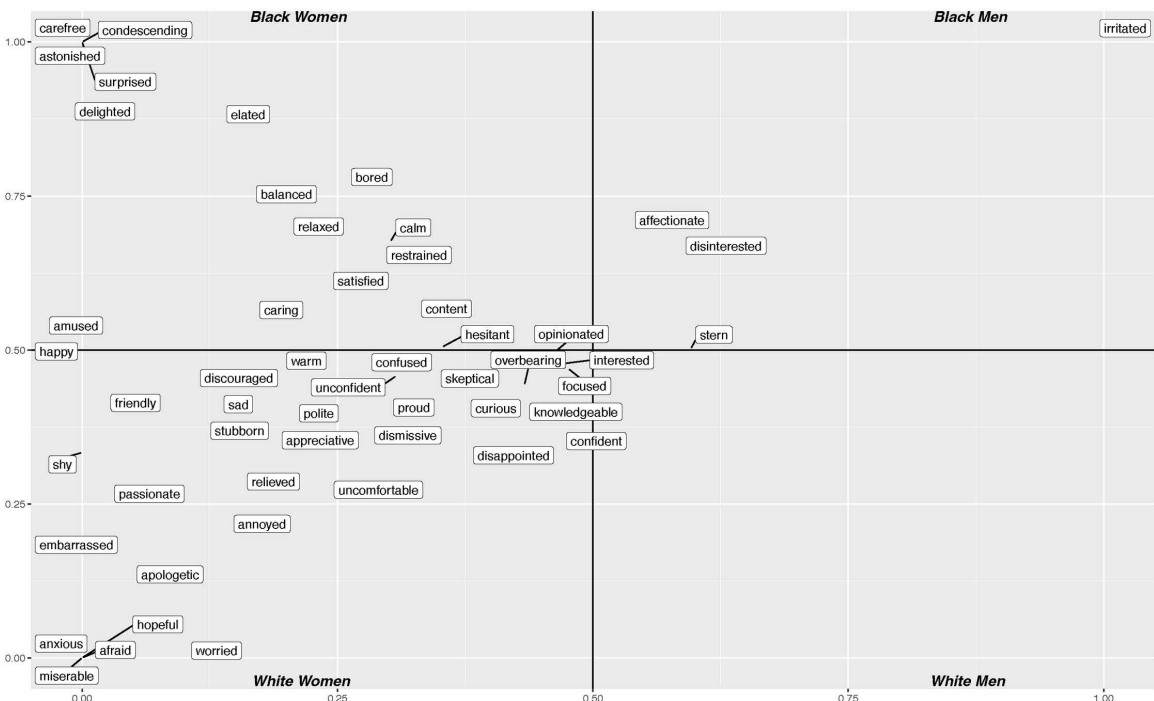

**Fig 3. Word cloud of descriptors from Halo.** Each quadrant is labeled for most frequent descriptors for that group.

for utterances from white men, and 22,101 for utterances from white women. However, as one moves closer to the corners, the descriptors are more likely to only be used only for utterances from speakers in that particular group.

For example, the descriptor 'irritated' is only used for Black men (129 times for Black men, 0 for the other groups), while only Black women's utterances were labeled as 'condescending', (51 times with 0 for the other groups). White women were the only group with utterances labeled 'miserable', 'afraid', or 'anxious' (780, 390, 591 times, respectively, with 0 for the other groups) As one can see, white men are the only group that does not have unique descriptors, suggesting that they were represented more broadly in the device's training data since they received a wider variety of adjectives.

## Discussion

In an analysis of the Amazon Halo's speech evaluation, we find differences between how the device evaluates speakers by race and gender, with women and Black speakers receiving lower Energy scores and more unique, negatively valenced descriptive adjectives. We also observe small differences with respect to Positivity scores, though results suggest that these are likely not derived from acoustic analysis of the speaker, but rather a speech-to text system that employed sentiment analysis, given the strikingly similar scores for all speakers. Overall, our findings support previous work by Harrington et al [18] and Yeung et al [17], which find that AI-based speech evaluation systems display systematic bias against marginalized groups. While Amazon discontinued support for the Halo device in 2023, these findings pose serious concerns for emerging AI-based conversation evaluation systems such as Read AI and the Zoom Revenue Accelerator, as well as emerging SFST systems designed for speech therapy and evaluations of mental health.

We find that the Halo device systematically provides lower Energy scores and more unique and more negative descriptive adjectives for women and Black speakers, perpetuating issues of algorithmic bias against speakers from marginalized groups. Even in a tightly-controlled reading task where content and context are not variable, the Halo's output reflects pervasive social biases in the determination of both scores and assignment of "tone" adjectives. This is particularly of note due to the fact that white, male speakers display less variation in their quantitative scores and receive no unique descriptive adjectives in our analysis, indicating that the system evaluates their speech as more normative with respect to its training data. Additionally, if these biases are a result of limitations in the Halo's training data, it is possible that they are reproduced or amplified for other marginalized groups with less representation in the training data, such as L2 English speakers or neurodivergent individuals. Though the current study controlled semantic content and speaker region, race, and gender, it did not consider other speaker demographic factors such as age or sexuality, which should be included in future research.

Though the Halo is one example, other SFSTs based on similar LLMs technologies are already being employed across domains such as speech therapy, language education, AI-based translation, and employment evaluation. Because LLMs use massive training data sets of ambiguous provenance and structure, the biases we observed here are likely to also emerge in any system that uses LLMs for the evaluation of speech. Firms such as Ambiq already market similar voice recognition technologies to diagnose depression, and the Zoom Revenue Accelerator uses analogous procedures to provide employment evaluations of employees to corporations [11]. In these contexts, pervasively lower ratings for one gender or racial group due to algorithmic bias can directly lead to systematic discrimination. Without careful attention to underlying issues in training data that lead to unequal evaluations by SFSTs, firms run the risk of embedding and amplifying social biases across domains [19].

These findings therefore demonstrate the need for careful consideration and testing of consumer-facing devices and other SFSTs that claim to evaluate "tone of voice", or to provide listeners information about how they sound to human listeners, particularly in domains such as healthcare and employment. Companies should proactively test their SFSTs programs and devices for such biases, and take steps to reduce them in their models and/or make users aware of the training and use cases for such systems. This could include providing transparent information about how specific systems may not function effectively for speakers from diverse backgrounds if they were not represented in the training data. Additionally, due to the fact that LLM-based systems often reproduce racist ideologies across societies even when firms attempt to explicitly adjust for them, companies should also interrogate whether the potential individual and social benefits that can result from the use of SFSTs in particular domains do indeed outweigh their potential harms [19]. The current moment provides the opportunity to proactively improve functionality and mitigate the potential harms of such technologies.

## Methods and materials

### Participants

Four participant groups - Black men, Black women, white men, white women - were recruited via Prolific (www.prolific.com) to record the reading passages described below. We recruited 17 Black women, 10 Black men, 16 white women, 15 white men, for a total of 58 participants. To mitigate regional variation, we utilized the ability within Prolific to focus recruiting geographically. All speakers were from the Midwest of the United States, to avoid certain regional linguistic features.

## Materials

Three reading passages were recorded – *The Rainbow Passage* [20], *Arthur the Rat* [21], and *Comma Gets a Cure* [22]. Each of these passages has been widely used in linguistic research. The online instructions were to read the passages in a quiet environment in the participant's home. Participants used computers and/or mobile devices to record. Given that the Amazon Halo device is to be used in a variety of settings, from quiet situations to loud environments with multiple talkers with varying SNR levels, the recording situations could be less strict than other experimental situations.

## Procedure details

Prior to playing the recordings, the Halo mobile app and the Amazon Halo device were calibrated following the instructions provided by Amazon. The mobile application was on a cell phone connected to the Amazon Halo via a Bluetooth connection. The Amazon Halo device was calibrated for the voice of the recording prior to each data collection. This procedure involves 'training' the device on the voice it will 'hear'. To do this, short portions of the recordings were played to familiarize the device with the current target speaker. After this calibration, the device was set to measure and score a Tone session. Then, the full reading passage recording from a single participant was played into the Amazon Halo device via a small loudspeaker in a sound-treated booth at 70 dB (as measured by a calibrated sound pressure level application in a mobile device). The Halo device was worn at the wrist of the investigator located approximately 18in from the speaker to model the approximate distance of the mouth from the device during a normal seated conversation.

Upon concluding the recording, the session was then processed on the mobile application. Amazon claims that all voice data was processed locally. The session's processing time varied, but the mobile application would then display an overall session score (i.e., highest Positivity, qualitative labels, etc.) The full session data was downloaded from Amazon. The Tone data includes information about the number of utterances from that session, the duration of the scored utterance, the date and time of the utterance and the session, various qualitative description labels for each utterance (e.g., competent, knowledgeable, tentative, angry, etc.), and our variables of interest - the score of Energy and Positivity for each utterance.

## Acoustic analysis

To obtain acoustic parameters that we would then statistically analyze, the recordings needed to be orthographically transcribed. The entire original recordings were transcribed and also divided into utterances according to the utterance durations provided by the data from the Tone session. What was immediately striking was that the durations of the utterances from the sessions were highly variable, from quite short (less than a second) to surprisingly long (more than 60 seconds). They did not always correspond to sentences, utterances, or breath groups. Attempting to discern how the utterances were divided by the Halo device proved to be impossible. Nonetheless, the investigators used the durations of the session utterances from the Tone session to segment the original recordings. Research on English has shown that one of the characteristics of its suprasegmental structure is phrases (see [23] for an overview, and [24] for a more in-depth discussion). Speakers group words into phrases, using prosodic phenomena (primarily using pitch). These phrases convey various intents and linguistic and social meanings, e.g., questions, emotions, etc. Listeners use them to interpret an utterance and also to gather social and emotional information about the speaker [25], e.g., is a response

needed, are they angry, what are some aspects of the person speaking. Without patterned phrasing, interpretation of any utterance - especially the emotional intent and/or the emotional valence of an utterance, becomes almost impossible. Nonetheless, using these transcribed and segmented utterances, all of the recordings were force-aligned to the phoneme level using the DARLA web interface [26]. We then used Praat Sauce [27] to extract 37 acoustic measures for each voiced phoneme from these force-aligned files. All voiceless phonemes were excluded.

The 37 acoustic measures evaluated were F0, F1, F2, F3, B1, B2, B3, H1u, H2u, H4u, H2Ku, H5Ku, A1u, A2u, A3u, H1H2u, H2H4u, H1A1u, H1A2u, H1A3uH2KH5Ku, H1c, H2c, H4c, A1c, A2c, A3c, H1H2c, H2H4c, H1A1c, H1A2c, H1A3c, CPP, HNR05, HNR15, HNR25, and HNR35.

## Table of acoustic measures

| Measures | Meaning | Linguistic Aspect |
|---|---|---|
| F0 | Fundamental frequency | One of the physical correlates of pitch |
| F1, F2, F3 | Formants of voiced sounds | Based on tongue position and other gestures of the vocal tract |
| B1, B2, B3 | Bandwiths of formants | Spread of acoustic energy of the various formants |
| H1u, H2u, H4u, H2Ku, H5Ku | Harmonics | Areas of increased energy based on properties of the vocal tract |
| A1u, A2u, A3u | Harmonics nearest the corresponding formant | Same as harmonics description above, but focused on those harmonics nearest a formant |
| H1H2u, H2H4u, H1A1u, H1A2u, H1A3uH2KH5Ku | Harmonic minus the following harmonic | A method of determining certain voice quality measures |
| H1c, H2c, H4c, A1c, A2c, A3c, H1H2c, H2H4c, H1A1c, H1A2c, H1A3c | Corrected versions of the above measures | Method used to normalize the measures to allow comparison across segment and person |
| CPP | Cepstral Peak Prominence | Measure of voice quality (harshness/roughness) |
| HNR05, HNR15, HNR25, and HNR35 | Harmonic to Noise Ratio | Measure of voice quality (breathiness) |

## Statistical analysis

Given the large number of acoustic measures and the lack of any previous literature to help point to what acoustic parameters that the Amazon Halo might use to provide the Positivity and/or Energy score, we included all of the 37 parameters provided by Praat Sauce. To aid in variable selection, and also to help reduce dimensionality, we decided to use LASSO regression modeling for our analysis. We fit two LASSO regression models - one with the outcome variable of Positivity and one with the outcome of Energy, both with all acoustic measures as potential predictors. LASSO (least absolute shrinkage and selection operator) regression is a tool for data sets with large numbers of possible predictors. LASSO regressions allow for high dimensionality data (such as ours) and helps to reduce multicollinearity (and overfitting of data). The regression introduces a penalty term in the residual sum of squares that is then multiplied by a regularization parameter. As the regularization parameter gets larger, the penalty also gets larger, which then reduces the impact of some of the model features. If the regularization parameter grows large enough, it can eliminate some model features, and serves as a type of automatic feature selection.

## Author contributions

**Conceptualization:** Nicole R. Holliday, Paul E. Reed.

**Data curation:** Nicole R. Holliday.

**Investigation:** Nicole R. Holliday, Paul E. Reed.

**Methodology:** Nicole R. Holliday, Paul E. Reed.

**Project administration:** Nicole R. Holliday.

**Resources:** Nicole R. Holliday.

**Validation:** Nicole R. Holliday.

**Visualization:** Nicole R. Holliday, Paul E. Reed.

**Writing – original draft:** Nicole R. Holliday, Paul E. Reed.

**Writing – review & editing:** Nicole R. Holliday, Paul E. Reed.

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
