## [Decision Letter · Decision Letter 0]

11 Sep 2024

PONE-D-24-23257

Gender and Racial Bias Issues in a Commercial “Tone of Voice” Analysis System

PLOS ONE

Dear Dr. Holliday,

Thank you for submitting your manuscript to PLOS ONE. After careful consideration, we feel that it has merit but does not fully meet PLOS ONE’s publication criteria as it currently stands. Therefore, we invite you to submit a revised version of the manuscript that addresses the points raised during the review process.

We look forward to receiving your revised manuscript.

Kind regards,

Michael Döllinger, Ph.D.

Academic Editor

PLOS ONE

Journal Requirements:

   "No"

3. In this instance it seems there may be acceptable restrictions in place that prevent the public sharing of your minimal data. However, in line with our goal of ensuring long-term data availability to all interested researchers, PLOS’ Data Policy states that authors cannot be the sole named individuals responsible for ensuring data access (http://journals.plos.org/plosone/s/data-availability#loc-acceptable-data-sharing-methods).

Reviewers' comments:

Reviewer's Responses to Questions

**Comments to the Author**

1. Is the manuscript technically sound, and do the data support the conclusions?

Reviewer #1: Yes

Reviewer #2: Partly

2. Has the statistical analysis been performed appropriately and rigorously? 

Reviewer #1: Yes

Reviewer #2: I Don't Know

3. Have the authors made all data underlying the findings in their manuscript fully available?

Reviewer #1: Yes

Reviewer #2: No

4. Is the manuscript presented in an intelligible fashion and written in standard English?

Reviewer #1: Yes

Reviewer #2: Yes

5. Review Comments to the Author

Reviewer #1: I fully acknowledge that I am not an expert in the software or algorithms behind the feedback speech technologies of the kind examined in this paper. But I think the article would be a better contribution if the authors said more about why the bias that they demonstrate is a significant detriment to well-being. "Employment evaluation" is tossed off in the last sentence as an afterthought. Was Halo used in this way? The only real explanation of the connection to health or well-being is the one the authors cite from the manufacturer, but there is no analysis as to whether this explanation corresponds at all to the reality of the use of Halo. Why, practically, would a Black woman, for example, be particularly concerned about the bias demonstrated here?

In addition, the authors focus on a product that no longer exists but then conclude in a sweeping way that their findings are pertinent for a range of products still in use. That statement should be explained. Are all the algorithms associated with these products so similar?

Minor points:

--The term "L2 English speaker" should be defined.

--There is inconsistent capitalization of "white" and capitalizations of terms such as "race" and "gender" that do not warrant capitalization.

--There is incorrect hyphenization of "the speaker side of the..." and of "speech signal" when they are used as nouns and not compound adjectives, but the hyphen is missing in the compound adjective "context-dependent", for example.

Reviewer #2: Thank you for this work. it is important to rigorously assess SFST and provide information about the basis of the manufacturer claims and potential bias. The work was carefully completed. The procedures and results sections would be clearer (and it would be more evident how the results support the discussion) with a few modifications.

1. It isn't clear what questions were being asked (or hypotheses tested - either frame would help)

2. Clarify the purpose of the acoustic analysis, provide a table with the names of the measures completed (in addition to abbreviations), the purpose of segmenting the utterances (and why it matters that you could not determine how HALO divided them).

3. Explain the regression + t-test decisions. What were you testing with each?

4. The presentation of the results did not seem to address all of the questions it appeared you were asking.

5. Because the results of the regressions were not quite clear and the values so close, the discussion seemed more definite than the data would lead one to see.

6. Side note: Figure 1 was not legible on either my monitor or when printed.

6. PLOS authors have the option to publish the peer review history of their article (what does this mean? ). If published, this will include your full peer review and any attached files.

**Do you want your identity to be public for this peer review?** For information about this choice, including consent withdrawal, please see our Privacy Policy .

Reviewer #1: **Yes: ** Joanne Csete

Reviewer #2: No

---

## [Author Response · Author response to Decision Letter 0]

10 Oct 2024

Please see attached document with reponses.

---

## [Editor Report · Decision Letter 1]

28 Oct 2024

PONE-D-24-23257R1Gender and Racial Bias Issues in a Commercial “Tone of Voice” Analysis SystemPLOS ONE

Dear Dr. Holliday,

Thank you for submitting your manuscript to PLOS ONE. After careful consideration, we feel that it has merit but does not fully meet PLOS ONE’s publication criteria as it currently stands. Therefore, we invite you to submit a revised version of the manuscript that addresses the points raised during the review process. *Comments from the Editorial Office: * Thank you for responding to the Academic Editor's previous comments regarding this submission. However, for the purposes of the PLOS ONE peer review process and confidentiality policy, we are issuing this decision directly to allow you to respond with an attached revised manuscript and response to reviewers file to the comments below: 

- "Even in a tightly-controlled reading task where content and context are not variable, the Halo’s algorithm implicitly utilizes race and gender information in the determination of its scores and assignment of “tone” adjectives" - while we agree the AI could have been trained using data from a specific gender and race and hence creating a bias, this statement above appears unsupported and appears to imply that the algorithm is using race and gender information to determine scores, which has not yet been shown. As such, we would recommend rephrasing/removing this statement.

- "These results also indicate that the observed biases may be even stronger in the types of naturalistic conversation that the Halo was designed to evaluate" - this statement does not appear directly supported by the results obtained.

- We recommend further discussion of the limitations of the study, including the small sample size as well as the fact that there could be differences in LLMs used in other devices, as this only tested the Halo.

- finally, the authors have cited the work of Hoffman et al 2024 in the body of the submission, but this is missing from the References section.

We look forward to receiving your revised manuscript.

Kind regards,

Avanti Dey, PhD

Senior Editor

PLOS ONE 

on behalf of

Michael Döllinger, Ph.D.

Academic Editor

PLOS ONE
---

## [Author Response · Author response to Decision Letter 1]

6 Nov 2024

Please see attached doc for revision reponse.

---

## [Editor Report · Decision Letter 2]

12 Nov 2024

Gender and Racial Bias Issues in a Commercial “Tone of Voice” Analysis System

PONE-D-24-23257R2

Dear Dr. Holliday,

We’re pleased to inform you that your manuscript has been judged scientifically suitable for publication and will be formally accepted for publication once it meets all outstanding technical requirements.

Kind regards,

Michael Döllinger, Ph.D.

Academic Editor

PLOS ONE
---

## [Editor Report · Acceptance letter]

PONE-D-24-23257R2

PLOS ONE

Dear Dr. Holliday,

I'm pleased to inform you that your manuscript has been deemed suitable for publication in PLOS ONE. Congratulations! Your manuscript is now being handed over to our production team.

Kind regards,

on behalf of

Dr. Michael Döllinger

Academic Editor

PLOS ONE